# Hashimoto Encephalopathy—Still More Questions than Answers

**DOI:** 10.3390/cells11182873

**Published:** 2022-09-14

**Authors:** Marta Waliszewska-Prosół, Maria Ejma

**Affiliations:** Department of Neurology, Wroclaw Medical University, 50-367 Wroclaw, Poland

**Keywords:** Hashimoto’s thyroiditis, Hashimoto encephalopathy, SREAT, autoimmunity, autoimmune encephalopathy, anti-thyroid antibodies

## Abstract

The normal function of the nervous system is conditioned by the undisturbed function of the thyroid gland and its hormones. Comprehensive clinical manifestations, including neurological disorders in Hashimoto’s thyroiditis, have long been understood and, in recent years, attention has been paid to neurological symptoms in euthyroid patients. Hashimoto encephalopathy is a controversial and poorly understood disease entity and the pathogenesis of the condition remains unclear. We still derive our understanding of this condition from case reports, but on the basis of these, a clear clinical picture of this entity can be proposed. Based on a review of the recent literature, the authors present the current view on the subject, discuss controversies and questions that still remain unanswered, as well as ongoing research in this area and the results of our own work in patients with Hashimoto’s thyroiditis.

## 1. Introduction

Thyroid hormones play a key role in the development and normal functioning of the nervous system. Their deficiency, in hypothyroidism, or excess, in hyperthyroidism, can cause a range of neurological deficit symptoms. Damage to the nervous system has also been observed in the course of autoimmune thyroid diseases in subjects with compensated thyroid function who have remained in euthyroidism for many years [1,2,3].

Currently, the most common autoimmune disorder in humans is Hashimoto’s thyroiditis (HT), also referred to as “chronic autoimmune thyroiditis” or “chronic lymphocytic thyroiditis”. The main feature of HT is lymphocytic infiltration of the thyroid gland, which on ultrasound examination shows a “*moth-eaten*” thyroid gland. Other characteristic features are the presence of serum anti-thyroglobulin (TGAb) and anti-thyroid peroxidase (TPOAb) antibodies and transient hormonal disturbances [4]. HT can present with euthyroidism and hypothyroidism, as well as the increasingly recently described hyperthyroidism [4,5].

Hashimoto encephalopathy (HE) is a controversial and poorly understood disease entity. Endocrinological and neurological bodies have repeatedly debated whether it is a myth or whether it really exists and has been given different names. Thus far, no consensus has been reached as to diagnostic criteria or management. Currently, our understanding of HE is based mainly on case reports or case series. The first time a patient with HE was described was by Lord Brain of Eynsham in 1966. He reported a 49-year-old man with Hashimoto’s thyroiditis who presented with recurrent cognitive dysfunction, impaired consciousness, hallucinations and stroke-like incidents from different regions of the brain vasculature. The symptoms receded after administration of corticosteroids [6]. Since this first report, there have been more than 300 reports of similar cases in the literature, of which more than 30 have been in children [7]. 

Many authors believe that a nomenclaturally more correct term for Hashimoto encephalopathy is SREAT—steroid-responsive encephalopathy associated with autoimmune thyroiditis. This term encompasses the possibility of encephalopathy occurrence also in the course of Graves–Basedow disease. In the literature, one can also find the abbreviation EAATD—encephalopathy associated with autoimmune thyroid disease—which is used interchangeably with SREAT [4,8,9]. Caselli et al. proposed the term NAIM—nonvasculitic autoimmune inflammatory meningoencephalitis [10].

This article describes the current state of knowledge regarding the thyroid–brain axis and the etiology, pathogenesis and controversy surrounding Hashimoto encephalopathy, which is expected to be the most serious neurological complication in Hashimoto’s thyroiditis.

## 2. Materials and Methods

### 2.1. Search Strategy

To prepare this manuscript, the search engines PubMed and Google Scholar were used, together with the following keywords: autoimmune thyroiditis, Hashimoto encephalopathy, SREAT, nervous system, brain, function, development, thyroid hormones, autoimmune encephalopathy and autoimmunity. In addition to using individual key words, to find the most relevant records the authors also used PubMed Advanced Search Builder. The advanced queries adopted were as follows: (((autoimmune) AND (thyroiditis)) AND (Hashimoto)) AND (encephalopathy); ((Hashimoto) OR (encephalopathy)) AND (((nervous system) OR (brain)) AND ((Hashimoto’s thyroiditis) OR (autoimmune)) AND ((SREAT) AND (EAATD) AND (NAIM) AND (thyroid hormones) AND (autoimmune encephalopathy))). To find the most relevant papers, two analysts working separately screened the search engine results.

### 2.2. Data Extraction

Further, 2178 records were identified and screened separately by the authors. Each of the analysts prepared their own list of records identified as being relevant to the study. Then, these record lists were double read by both analysts and 923 abstracts were found to be relevant to the subject. Next, full-text manuscripts were acquired. All articles were read independently by both analysts and duplicate publications were removed. Reviews and research studies, classified according to their relevance, were initially included, with subsequent exclusion of conference abstracts and papers written in languages other than English.

### 2.3. Qualitive Analysis and Synthesis

Each analyst worked independently and prepared their own list of relevant full-text manuscripts. Both lists were compared and 114 publications were found to be the most relevant to the study and included in this review.

## 3. Results

### 3.1. Thyroid and Brain

In fetal life, thyroid hormones have a significant influence on growth processes and the formation of cytoarchitectonics of the central nervous system (CNS). They participate in neurogenesis, differentiation of neurons and glial cells, formation of the six-layer cortex, myelinization and migration of neurons in the cortex, hippocampus and cerebellum [11,12]. They play an important role in synaptogenesis as well as in the regulation of GABA- and noradrenergic conduction [13]. It has been shown that thyroid hormone deficiency in fetal life can result in numerous structural defects of the CNS. In congenital hypothyroidism, there is a decrease in the number of cells in the olfactory bulb, granular layer of the CA1 and CA4 region in the hippocampus and cerebellum and loss of GABA-ergic neurons in the cerebellum and interneurons in the cerebral cortex [13,14,15]. There may also be neuropil reduction, deformation or permanent structural damage to dendritic cells or Purkyni cells in the cerebellum and cholinergic and serotonergic neurons [12,15,16,17].

Since the beginning of research into mechanisms of attention and memory, thyroid hormones have attracted the attention of researchers due to the numerous clinical observations of impaired learning in hypothyroidism and accelerated associative thinking in hyperthyroidism. It is now known that thyroid hormones, along with retinoids, are key signals regulating neuronal plasticity associated with learning [18,19]. 

More than 1100 thyroid target genes have also been shown to be essential for the brain [17]. Among the most important thyroid hormone-regulated genes that influence brain plasticity, memory and learning processes are the protein kinase C substrate, *NRGN* (RC3/neurogranin), the amyloid precursor protein gene producing β-amyloid precursor (Aβ), brain-derived neurotrophic factor (BDNF) and the expression of neuromodulin/GAP-43 [20,21,22]. 

Thyroid-dependent apoptosis processes lead to a reduction in neuronal networks important for brain maturation, as well as programmed cell death in the brain [23]. Thus, it seems likely that one of the major causes of early dementia in hypothyroidism is a disturbance of programmed neuronal population formation during brain development. Moreover, thyroid hormones control the fate of brain stem cells and, through the activation of Ca2+-ATPase, K+-Na+ -ATPase, αvβ3 integrin and Gq signaling mitogen-activated protein kinases, may affect bioelectrical processes in neural tissue [19,20,24].

The proper functioning of the CNS after maturity is also dependent on thyroid hormone levels. During this period, proper T3 and T4 levels are ensured by effectively functioning regulatory mechanisms, including adequate thyroid gland secretion, proper deiodinase expression and unimpaired thyroid hormone transport to the CNS, especially across the blood–brain barrier [12,13,25].

### 3.2. Hashimoto’s Thyroiditis

Hashimoto’s thyroiditis (HT) is the most common endocrine disorder and the most common cause of primary hypothyroidism in both children and adults [4,26]. The highest incidence is observed between 40 and 65 years of age. Women suffer from the disease 10–20-times more often than men, which may indicate the involvement of female sex hormones in the pathomechanism of inflammation. HT is estimated to occur in 1–2% of the general population, more often in Caucasians and Asians than African Americans. This figure is most likely underestimated, however, because autopsy studies reveal characteristic lymphocytic infiltrates in the thyroid gland in up to 20–40% of subjects [27]. Additionally, 25% of adult women and 10% of men and more than 30% of women over 70 years of age who have never had thyroid disease are found to have high titers of antithyroid antibodies [4,27,28]. In contrast, Mariotti et al. [29] showed that in a population of individuals aged 100–108 years, anti-TPO antibodies were present in only 5.8%. They suggested a longer survival of individuals in whom these antibodies were not produced.

Hashimoto’s thyroiditis may be part of autoimmune polyendocrine syndromes (APS), which also include type I diabetes, Addison–Biermer disease, rheumatoid arthritis, coeliac disease, vitiligo or Sjögren’s syndrome [28,30,31]. It may also coexist with autoimmune neurological disorders—myasthenia gravis, multiple sclerosis or chronic inflammatory demyelinating polyradiculoneuropathy [32,33]. Clinical observations of complex autoimmune syndromes allow us to assume the existence of a common genetic background; however, the pathomechanism of these diseases has not been fully explained [34]. 

HT is characterized by the presence of serum anti-thyroglobulin (TGAb) and anti-thyroperoxidase (TPOAb) antibodies and this is one of the diagnostic criteria for the diagnosis of this disease [28]. However, over the last few years, it has been shown that these are not the only antibodies directed against thyroid cells. Other anti-thyroid immunoglobulins may also be detected in patients with autoimmune thyroiditis: TSH receptor blocking or stimulating antibodies (TSHRAb), antibodies to thyroxine, triiodothyronine, megalin (a transmembrane protein belonging to the low-density lipoprotein receptor family), pendrin (anion transport protein SLC26, carrying iodide ions to the colloid) or against sodium-iodide symporters (anion transport protein SLC5A catalyzes the active transport of iodide ions from the blood into the thyroid). Anti-pendrin antibodies are found in up to 97% of patients with Hashimoto’s thyroiditis and this may one day be used in the diagnosis of the disease. The exact role of these antibodies in the pathogenesis of the disease has not yet been determined and they have no diagnostic significance at present [4,35,36]. It should also be added that antithyroid antibodies are also found in other thyroid diseases (e.g., neoplasms, Graves–Basedow disease), in healthy individuals without thyroid dysfunction, as well as in patients with other autoimmune diseases (e.g., rheumatism) [37].

The mechanisms that trigger the immune response directed against thyroid structures have not yet been completely elucidated. However, there is no doubt that the pathogenesis of HT is complex and multifactorial. Both a specific genetic susceptibility and the harmful effects of certain environmental factors are important in its development [4,28,37].

In the etiopathogenesis of HT, an abnormal immune response plays a fundamental role, resulting from genetic susceptibility and environmental conditions [38]. The disease causing the development of autoimmune thyroiditis proceeds over several stages (Figure 1).

The first, preliminary and crucial, is the activation of non-sensitized CD4+ T lymphocytes. T lymphocytes mainly identify autoantigens with a protein structure. Appropriately prepared peptides (e.g., by proteolytic fragmentation) are presented to T lymphocytes with the participation of “professional” antigen-presenting cells (APC) and, in the case of HT, “non-professional” APC cells—thyrocytes. Peptides presented in communication with MHC class II molecules are recognized by Th lymphocytes. The prerequisite for T lymphocyte activation is the binding of the T cell receptor (TCR) (*signal 1*) and the interaction of CD80 and CD86 molecules on the APC with CD28 molecules on the T lymphocyte (*signal 2*) [39,40]. The absence of signal 2 causes the lymphocyte to become anergic. For autoimmunity not to occur, autoreactive T cells must undergo elimination or remain anergic. When a Th lymphocyte is activated, a series of biochemical changes is triggered, resulting in the release of various cytokines (e.g., interleukins, interferon γ or TNF-α), which have the ability to modulate the immune response. The local presence of leukocytes is also required for the proper development of the immune response. Adequate migration is ensured by the chemokine system. Recent studies indicate that chemokines CXCL9, CXCL10 and CXCL11, induced by INF-γ, may play an important role in the early development of Hashimoto’s thyroiditis [39,40,41].

Subsequent stages of disease progression follow the breaking down of natural peripheral and central tolerance to one’s own antigens. In a healthy person, peripheral (late) tolerance occurs in the peripheral lymphoid organs (e.g., lymph nodes or spleen) and involves elimination, blocking the action or preventing the stimulation of lymphocytes. Central (primary) tolerance is associated with the elimination of autotreated lymphocytes at the stage of their maturation in the central lymphoid organs—thymus, bone marrow and fetal liver. One of the main mechanisms of self-tolerance is the phenomenon of anergy. However, animal studies have shown that local production of IL-2 or infecting mice with nematodes can activate previously anergic lymphocytes [38,42,43,44]. Another reason for the breakdown of immune tolerance may be the deficiency and/or malfunction of a certain subpopulation of lymphocytes, called regulatory T cells (Treg). These cells are involved in the control and suppression of excessive immune responses, especially against one’s own antigens. They mainly include CD4+CD25+ lymphocytes (CD4 with constant expression of the receptor for the α chain of IL-2), which suppress the function of CD4 cells, CD8 cells, macrophages, dendritic cells and natural killer (NK) cells [29]. Treg lymphocytes also include CD8+CD122+ cells, type 1 regulatory cells (Treg-1), which secrete IL-10 and TGF-β and type 3 helper cells (Th3), which mainly secrete TGF-β [40,45].

In the next stage of disease development, cytotoxic CD8+ lymphocytes and B lymphocytes are activated and transform into plasma cells. Plasmocytes produce and secrete antibodies against various thyroid antigens, mainly against TPO, Tg and the TSH receptor. Th1, Th2 and Th17 lymphocytes are involved in the pathogenesis of HT [46,47]. The largest pathological contribution is attributed to a subpopulation of Th1 lymphocytes, which is associated with a cell-type response. Th1 lymphocytes produce the pro-inflammatory cytokines IL-2, IL-6, TNF-α, IFN-γ and secondarily activate macrophages. Th2 cells participate in the humoral response and are responsible for stimulating B lymphocytes, which produce antibodies. They secrete IL-4, IL-5, IL-6, IL-10 and IL-13 and can inhibit the production of pro-inflammatory cytokines by affecting Th1 cells; in addition, they activate anti-apoptotic molecules [47,48]. A subpopulation of Th17 lymphocytes mainly produces the cytokines IL-17, IL-21 and IL-22, regulates inflammatory processes and participates in the response directed against microorganisms and in the development of allergic diseases. The association of Th1 and Th17 cell infiltration in the thyroid gland with apoptosis and the inflammatory process of this gland has been confirmed in experimental models in mice [38,48]. Bossowski et al. [46] showed an increase in IL-17 levels in children with newly diagnosed Hashimoto’s disease, which also suggests the involvement of Th17 in its pathogenesis.

The next stage of the disease is the development of lymphocytic thyroiditis, accompanied by the production of antithyroid antibodies. A sensitive indicator of the autoimmune process is anti-TPO antibodies found in almost all patients [49]. These are a heterogeneous group of about 180 types of antibodies, mostly of the IgG class, which recognize different epitopes on the molecule of peroxidase, the basic enzyme of thyroid hormone synthesis. These antibodies are responsible for the formation of complement system-dependent cytotoxicity and antibody-dependent cellular cytotoxicity [38,50]. Xie et al. [51] distinguished four subclasses of anti-TPO antibodies occurring at different frequencies in patients with Hashimoto’s thyroiditis. IgG1 subclass antibodies were observed in 70.2% of patients, IgG2 in 35.1%, IgG3 in 19.6% and IgG4 in 66.1%. The different subclasses have been shown to have different biological functions and their distribution can have important clinical implications. Anti-TPO antibodies in the IgG4 class are particularly important in the pathogenesis of the disease and influence the clinical picture. These antibodies usually reach high levels in serum and their presence is associated with male gender, young age at onset, rapid destruction and fibrosis of the thyroid gland and development of its hypothyroidism (overt or subclinical). It has been observed that IgG4 antibodies are present in 70% of patients with hypothyroidism, compared to only 47% of euthyroid patients [51]. Some researchers suggest that HT should be divided into IgG4 positive and IgG4 negative, due to their different clinical presentations and prognoses [52,53]. A less important determinant of the disease is anti-TG antibodies, which can occur in isolation or coexist with anti-TPO antibodies. They can also appear in the serum of healthy individuals who never develop autoimmune thyroid disease. Nevertheless, observations indicate that their presence in relatives of people with Hashimoto’s thyroiditis is a marker of increased risk of the disease. Anti-TG antibodies are not involved in antibody-dependent cellular cytotoxicity and are not capable of binding complement [54]. 

The final stage of the immune process in HT is the destruction and fibrosis of the thyroid gland in the course of a progressive process of cell apoptosis. The lesions are the result of extensive infiltration, consisting of cytotoxic T lymphocytes, B cells and macrophages. Tyreocytes attacked by the complement system release pro-inflammatory molecules (prostaglandin E2, IL-1 and IL-6), which activate cytotoxic lymphocytes [38,40]. These, in turn, trigger the activation of intracellular caspases, which ultimately leads to programmed cell death [38,54,55]. In parallel with the morphological changes, the clinical picture shows initially subclinical hypothyroidism, which gradually progresses to overt hypothyroidism, as the thyroid gland is destroyed and fibrotic [39].

The diagnosis of HT is established on the basis of clinical presentation, thyroid hormone levels, the presence of serum anti-thyroid antibodies and ultrasound evaluation of the thyroid gland [28].

The clinical manifestations of the disease are often non-specific or modest. They are divided into local, resulting from the presence of goiter (e.g., dysphonia, dysphagia, dysphagia), and systemic, resulting from loss of thyroid function [56]. They are usually associated with subclinical or overt hypothyroidism, less commonly with hyperthyroidism (so-called hashitoxicosis). Given the wide influence of thyroid hormones on the function of many organs and tissues, the symptoms of the disease can be varied. The most important and most common include: increased body weight, obesity, fatigue, decreased exercise capacity, increased need for sleep, constant feeling of cold, slowed intestinal peristalsis, constipation, bradycardia, decreased cardiac output, enlarged heart silhouette, tendency to hypotension, menstrual disorders, impotence, miscarriages in the first trimester of pregnancy, impaired wound healing, dry skin, brittleness and hair loss [56,57]. Nervous system damage may manifest as neuropathy, myopathy and, less commonly, encephalopathy. Physical examination of the thyroid gland typically reveals features of a non-painful goiter with an uneven surface and increased consistency—less commonly, a large nodular or atrophic goiter [26,28,57].

### 3.3. Hashimoto Encephalopathy

Hashimoto encephalopathy suggests a pathogenetic link to Hashimoto’s thyroiditis, despite the fact that symptoms of similar encephalopathy have also been described in Graves’ disease. HE is one of the most elusive diseases and, despite more than 50 years of observation, its etiology and treatment have not been clearly established [58]. It has been called by different names according to different criteria. Still, the most popular term is Hashimoto encephalopathy. Many researchers suggest that reported cases of patients diagnosed with HE are, in fact, a “coincidence” and not a proper diagnosis and some believe that HE is the main representative of modern neuromythology [59].

#### 3.3.1. Epidemiology

Hashimoto encephalopathy is a very rare disease and its prevalence is estimated at 2.1/100,000 in the adult population [60]. HE has been described in all age groups, including children, and most commonly in the 4th–6th decades of life. The disease is more frequently observed in women than in men (4:1 ratio), which probably results from the fact that Hashimoto’s thyroiditis is more common in females [60,61]. In approximately 30% of described cases, HE coexisted with other autoimmune disorders, such as type I diabetes, systemic lupus erythematosus and Sjögren’s syndrome [62]. Despite the more frequent familial occurrence of Hashimoto’s thyroiditis, no cases of familial HE have been described. 

The incidence of HE is probably much higher. The reason for the low recognition of HE is probably the limited knowledge of this disease among physicians and the lack of routine antithyroid antibody testing—especially in patients with thyroid diseases and cognitive impairment and dementia [2,63,64]. 

#### 3.3.2. Pathogenesis

The pathogenesis of the disease is still unclear. Based on previous observations, it is known that hormonal disturbances probably do not play a role in the development of HE because thyroid function was usually balanced in the described cases [62,63]. The presence of anti-thyroid antibodies, effective corticosteroid treatment and the variable course of the disease with periods of exacerbation and remission suggest an autoimmune background. However, no correlation has been demonstrated between the level of antibodies and the severity of the disease or the constellation of clinical symptoms [6,8]. 

Experimental studies have also not confirmed any direct pathogenic effect of these antibodies (especially TPOAb) on the nervous system. Despite their often high concentrations in the blood serum and even in the cerebrospinal fluid, autopsy and histopathological examinations have not shown the presence of antibodies in the structures of the nervous system, but only in the tissues of the thyroid gland [61,63]. A positive correlation has been found between the anti-TPOAb titer and the degree of thyroid fibrosis in ultrasound images, which was not observed in the case of anti-TGAb, which probably have less influence on the destructive processes of the thyroid gland [65]. Therefore, for many researchers, the criterion of high titers of anti-thyroid antibodies in the diagnosis of HE is controversial. They emphasize that the concentration of anti-TPOAb and anti-TGAb in patients with Hashimoto’s thyroiditis may change over time and should be treated only as a marker of the ongoing autoimmune thyroid process. Moreover, researchers note that in patients diagnosed with HE, no decrease in antibody titers has been observed after corticosteroid treatment [2,4,65].

Morphological and histopathological studies in patients diagnosed with HE have shown the presence of chronic, limited inflammatory lesions of the cortex and meninges, which have been termed NAIM—*nonvasculitis autoimmune inflammatory meningoencephalitis* [10]. Some autopsy studies raise the suspicion of an association of HE with lymphocytic infiltration or vasculitic inflammatory lesions of the brainstem and gray matter [66]. 

Direct toxic effects of thyrotropin-releasing hormones (TRH) on nerve cells or inflammatory lesions of the brain and medulla in the course of demyelination have also been considered as causative factors. This hypothesis was proposed after a patient developed myoclonus and tremor after an intravenous TRH infusion [67].

It has also been suggested that HE may follow a generalized decrease in cerebral perfusion or brainstem cytotoxic edema. The microcirculatory abnormalities would be secondary to the deposition of immune complexes in the brain vessels, as demonstrated by the hypoperfusion on SPECT (single-photon emission computed tomography) described in case reports [68,69].

In recent years, attention has turned to an aggressive form of Hashimoto’s thyroiditis accompanied by elevated serum IgG4 levels. It is characterized by a higher incidence in men (5:1) than in women, onset at a younger age, more intense thyroid inflammation and higher antithyroid antibody titers [35,53,55]. Individuals with this form of Hashimoto’s thyroiditis have excessive production of IgG4+ plasmocytes, which infiltrate various organs leading to their fibrosis and sclerosis, sometimes resulting in inflammatory tumors [70]. This form of Hashimoto’s thyroiditis has been classified as IgG4-RD (IgG4-related disease). 

The first reports of IgG4-related disease processes appeared in the literature in 2001 and these concerned autoimmune pancreatitis. Currently, the IgG4-RD group includes a considerable number of autoimmune diseases, e.g., sclerosing cholangitis, interstitial nephritis, Sjögren’s syndrome, Riedel’s goiter or chronic thyroiditis [70,71,72]. Hosoi et al. [73] described a 60-year-old man with features of severe Hashimoto encephalopathy who had elevated IgG4 levels in both serum and cerebrospinal fluid. The patient’s IgG4 index was lower than that of IgG, indicating passive transport of IgG4 across the blood–brain barrier rather than primary synthesis of this immunoglobulin fraction in the CNS. After treatment with corticosteroids, serum IgG4 levels decreased but were indeterminate in the cerebrospinal fluid. The described case could suggest the involvement of the IgG4 fraction in the development of neurological disorders, including encephalopathy in patients with Hashimoto’s thyroiditis.

##### Autoantibodies against the Amino (NH2)-Terminal of α-Enolase (aNAE)

In recent years, a new antigen, α-enolase, has been discovered in the brains of Hashimoto encephalopathy patients and high titers of antibodies to α-enolase have been found in their serum and cerebrospinal fluid [74]. This was then considered a potential biomarker of HE [75].

Unfortunately, it was very quickly demonstrated that these antibodies are not specific. α-enolase is expressed not only in the blood vessels of the brain (including in embryonic neurons) but in most cells throughout the body [76]. It has also been shown that elevated serum levels of autoantibodies to α-, γ- or both enolases are found in a variety of autoimmune diseases. A particular predisposition to the disclosure of these antibodies has been demonstrated in diseases that manifest with systemic vascular damage: rheumatoid arthritis, systemic lupus erythematosus, ANCA-positive vasculitis, autoimmune nephritis, primary Sjogren’s syndrome, systemic sclerosis, primary biliary cirrhosis, inflammatory bowel disease and celiac disease [76,77,78].

Analysis of the presence of α-enolase in patients with HE showed that they were demonstrated in 68% of patients compared to 12% of patients with autoimmune thyroiditis without encephalopathy. Moreover, they have also been identified in a patient with Creutzfeldt–Jakob disease and in patients with limbic encephalitis [79]. Matozzi et al. also demonstrated in a group of 24 patients with HE that aNAE is not specific: they were shown only in three patients with HE and one patient with another type of encephalopathy [80].

#### 3.3.3. Clinical Manifestations

The clinical manifestations of Hashimoto encephalopathy are not specific. The onset of the disease can be insidious and uncharacteristic and the course—usually acute or subacute—often includes relapses. Cognitive dysfunction has been described as the initial symptom in more than 80% of patients and behavioral and personality disorders in more than 90–100% [60,62,81].

Based on previous reports, HE is conventionally divided into two types (Table 1).

The first features recurrent incidents suggesting a vascular background of symptoms (stroke-like incidents, corresponding to transient cerebral ischemia), with frequent cognitive impairment but without epileptic seizures.

The second is progressive in nature, with epileptic seizures, disturbances of consciousness, psychiatric symptoms, such as mania, depression or psychosis, and an increasing dementia syndrome. The first type is the rarer, milder and most frequently relapsing form of HE; the second type is more common, usually with an acute onset and rapid course, usually running without relapses [7,61,82]. 

However, the spectrum of disorders observed in Hashimoto encephalopathy is very wide. In both forms of HE, tremor, myoclonus, ataxia, stupor or coma may occur [82,83,84]. Less frequently, persistent headaches, gait disturbances, isolated nystagmus, opsoclonus-myoclonus syndrome, parkinsonian syndromes, spastic paraparesis, fatigue syndrome, sleep disturbances or visual and auditory hallucinations have been reported [60,61]. Patients treated for many years for schizophrenia, depressive disorders, Alzheimer’s disease or drug-resistant epilepsy have been described, in whom HE was finally diagnosed and corticosteroid therapy resulted in complete resolution of symptoms [82,85]. On the other hand, when analyzing the relationship between psychiatric disorders and thyroid diseases, Hashimoto’s thyroiditis, even in euthyroid state, increases the risk of their occurrence [86]. Isolated studies indicate a correlation between anti-thyroid antibody levels and anxiety, dysthymia and somatization in patients with HT; however, others do not confirm this [87,88,89]. According to some researchers, the presence of anti-TPOAb in depressed patients is associated with a worse response to antidepressant treatment [84]. A relationship between HT and mood disorders is being sought in autoimmune processes. 

Seizures occur in approximately 60–70% of patients with HE [2,90]. They may take the form of generalized tonic–clonic seizures, which have been observed more frequently in children, focal simple or complex seizures, and also myoclonic seizures or the absence of seizures [91,92]. Despite a good prognosis, fatal cases of status epilepticus in the course of HE have been described [93,94]. 

However, it is important to keep in mind that epilepsy is more frequently observed in people with autoimmune disorders than in the general population [95]. This confirms the existence of cerebral cortex hyperactivity in patients with immunological disorders. Epileptic seizures are part of the clinical picture of systemic lupus etythematosus, sarcoidosis, Sjögren’s syndrome, coeliac disease, Behcet’s disease, Wegener’s granulomatosis as well as Hashimoto’s thyroiditis [95,96,97]. The pathogenesis of these seizures remains unclear. Possible causes that are being considered include autoimmune mechanisms directly affecting CNS structures (antibodies, circulating immune complexes, proinflammatory cytokines), alteration in cerebral vascular structures in the course of the inflammatory process, metabolic disorders or complications of therapy [97,98,99]. It has been shown that both children and adults with epilepsy are more likely to have various antibodies than the general population [97]. Tsai et al. [99] found elevated titers of antinuclear (ANA), antimicrosomal (AMA) and anti-TPO autoantibodies in 28.3% of patients with idiopathic epilepsy.

A rarer manifestation of HE is stroke-like incidents, usually fluctuating, which occur in about 25–30% of patients [62,100]. They have the character of hemispheric motor and sensory disturbances or other neurological deficits from different areas of brain vascularization. 

Interesting conclusions were provided by a recent study characterizing a group of 56 patients with transient global amnesia (TGA), which appears to be a disorder from the spectrum of vascular diseases and not, as had been previously thought, epileptic. Autoimmune thyroiditis was the only comorbidity in 17.9% of patients [101]. TGA is associated with disturbance of microcirculation in the hippocampus and in one patient with HE, MRI revealed bilateral hippocampal or thalamic lesions [102]. TGA might perhaps be considered an early manifestation of encephalopathy in the course of autoimmune thyroiditis. This requires further observation.

Isolated reports describe rare cases of HE, in which peripheral nervous system damage also coexisted [103,104,105]. In a 35-year-old woman with Hashimoto’s thyroiditis, Sheng et al. [105] observed flaccid lower limb paresis caused by demyelinating polyneuropathy with multiple conduction blocks in addition to cerebellar symptoms. In this case, brain MRI showed multiple inflammatory lesions in the brainstem, thalamus and cerebellum. Other authors have described HE coexisting with sensory ganglionopathy and painful neuralgic amyotrophy [104,105]. In all these cases, the neurological symptoms receded after immunosuppressive treatment. 

#### 3.3.4. Criteria for Diagnosis

The diagnostic criteria for HE are based on certain clinical features with elevated levels of antithyroid antibodies and also a good response to steroid treatment. The diagnostic criteria were proposed by Graus (Table 2) and suggest that the diagnosis of HE remains a diagnosis by exclusion [62,106,107,108].

The inclusion of ‘good response to corticosteroid treatment’ as a necessary criterion in the criteria remains controversial. Recent studies suggest that such a response is achieved in only 32% of patients with HE [80]. 

#### 3.3.5. Diagnostic Findings

There are usually no significant abnormalities in the basic laboratory tests. Inflammatory markers are negative and patients do not have fever. TSH levels are usually normal and the degree of thyroid function balance does not correlate with clinical status. In more than 70% of the cases described, patients were euthyroid or subclinically hypothyroid. All of them had elevated serum anti-TPOAb and/or anti-TGAb [4,82].

In more than 85% of patients, cerebrospinal fluid examination shows elevated protein levels, which normalize after treatment. Less frequently, in about 10–25% of cases, a slight lymphocytic pleocytosis is found. In the majority of patients (62–75%), antithyroid antibodies are found in the CSF [108]. Some patients might have other autoimmune antibodies, such as N-methyl-D-aspartate receptor (NMDAR), gamma-aminobutyric acid A receptor (GABAAR), contactin-associated protein 2 (Caspr2), LGi1, a-amino-3-hydroxy-5-methylisoxazole-4-propionic acid receptor 1 and 2 (AMPAR1/2) or antinuclear antibodies (ANA) [79,82,90,95]. Isolated cases have been described in which oligoclonal bands, 14-3-3 protein and anti-α-enolase antibodies were present in the fluid [69,74,79,109].

EEG abnormalities are found in approximately 82–98% of patients [2,90]. The abnormalities are non-specific: generalized basal activity decreases and paroxysmal delta and theta wave discharges have been reported [92]. The location of epileptic activity is not always consistent with the site of lesions shown on neuroimaging [90,92]. A single report described a man with HE, which, in its course, mimicked Creutzfeldt–Jakob disease [110]. The patient’s EEG showed characteristic synchronous discharges typical of a Radermecker recording. In most cases of Hashimoto encephalopathy, EEG changes recede after steroid treatment [95,110]. Some observations using multimodal evoked potentials (EP) confirm brain bioelectrical activity disturbances in Hashimoto’s thyroiditis patients who do not have central nervous system deficits. The increased amplitudes of the EP may indicate increased cerebral cortex activity, which may be associated with an ongoing autoimmune process [111]. Analyzing the event-related potentials (ERP), we showed significantly prolonged N200 and P300 waveform latencies from all leads. In correlation with magnetic resonance spectroscopy (MRS), the N-acetylaspartate/creatine (NAA/Cr) ratio showed significant negative correlations with all N200 latencies and the myo-inositol/creatine (mI/Cr) ratio showed a significant positive correlation with P300 latencies. This indicates that prolonged response latencies are associated with metabolic changes in the cerebral cortex [112].

Imaging studies usually do not show abnormalities, although in some cases, uncharacteristic changes are observed. The most frequently described abnormalities include small, scattered vascular foci in the white matter, changes suggestive of a demyelinating process of ADEM (acute disseminated encephalomyelitis) character or vasculitis and, less frequently, features of cytotoxic edema and cerebellar atrophy [63,113,114,115]. Forchetti et al. [68] described an improvement in cerebral flow seen on SPECT after effective treatment of HE.

Advanced neuroimaging techniques—MRS—show that in patients with compensated Hashimoto’s thyroiditis without symptoms of damage to the nervous system, there were disturbances in the metabolic composition of the brain: a significant decrease in the N-acetylaspartate/creatine (NAA/Cr) ratio in the posterior cingulate regions and in the white matter of the left parietal lobe. The reduction in NAA/Cr ratios may suggest a loss of neuronal activity within normal-appearing gray and white matter in patients with HT [116]. Diffuse leukoencephalopathy-like damage, limbic encephalitis and hippocampal edema have been reported in isolated cases [117,118].

#### 3.3.6. Differential Diagnosis

The differential diagnosis of Hashimoto encephalophaty should include all neurological and psychiatric diseases that can cause rapidly progressive dementia and that may clinically resemble the symptoms seen in HE. 

In this group of conditions, the following should be considered: meningeal and cerebral involvement of infectious, neoplastic or paraneoplastic origin, neurodegenerative diseases (Alzheimer’s disease, dementia with Lewy bodies, frontotemporal dementia), Creutzfeldt–Jakob disease, acute disseminated encephalomyelitis (ADEM), toxic, metabolic and autoimmune encephalopathies, carcinomatous meningitis, paraneoplastic encephalitis, psychiatric diseases (psychosis, mania, depression, anxiety disorders), vascular disorders, cerebrovasculitis, transient global amnesia (TGA) and basilar or hemiplegic migraine [7,95,101,117,118,119].

#### 3.3.7. Treatment and Prognosis

There are no clearly established treatment guidelines for Hashimoto encephalopathy. Corticosteroid therapy is the treatment of choice and the first-line drug is methylprednisolone 1 g i.v. for 5 days. Some authors suggest the necessity for subsequent oral steroid therapy. In that case, prednisone at an initial dose of 50–150 mg daily or 1–2 mg/kg/day is recommended [60,82,120]. 

In cases when corticosteroid treatment is not effective, treatment with methotrexate, azathioprine or cyclophosphamide should be considered. There are also reports on the effectiveness of treatment with plasmapheresis and immunoglobulins (IVIg). Rituximab has been successfully used in a patient with opsoclonus–myoclonus syndrome [82]. In patients who experience epileptic seizures, antiepileptic drugs should only be used ad hoc. Their efficacy in forms of HE with epileptic seizures has not been proven and seizure reduction has been observed only after immunosuppressive treatment. In recent years, T-cell inhibitors (tacrolimus, cyclosporine A and sirolimus) have been found to be successfully used to control refractory epileptic seizures. Other drugs, such as methotrexate, azathioprine and hydroxychloroquine, have also been shown to be effective in some cases [90,121,122]. Seizures can recur, especially when the dose of steroids is reduced, so maintenance immunotherapy is necessary in some patients [90,122]. In cases of subclinical or overt hypothyroidism, treatment with levothyroxine should be initiated.

In about 95–98% of patients after immunosuppressive therapy, the prognosis in HE is an improvement or complete disappearance of symptoms [123,124].

## 4. Controversies

Hashimoto encephalopathy continues to be extremely controversial, despite years of observation. Skeptics and opponents of this disease entity emphasize that many described cases of HE appeared before the expansion of the spectrum of autoimmune antibodies and only new developments in advanced laboratory techniques allow for the identification of the cause of autoimmune encephalopathies/encephalitides. When the first cases of HE were published, tests for other autoantibodies in serum or cerebrospinal fluid did not exist. In addition, we know that autoimmune diseases can coexist and that many different antibodies can be found in patients. In addition, anti-TPO antibodies specific for HE are generally not specific and may occur in up to 13% of the general population and in other autoimmune diseases without thyroid manifestation. Therefore, there is a suspicion that most of the described cases might have been underdiagnosed and the diagnosis of HE was completely coincidental, even though it was a diagnosis by exclusion.

However, despite this controversy, further follow-up, including a broad differential diagnosis, should be performed in view of the potential for cure. On the other hand, some researchers emphasize that the initiation of treatment with immunosuppressive drugs carries with it many potential side effects and should not be hastily adopted only on the basis of nonspecific clinical symptoms and elevated antibody titers common in the general population.

## 5. Conclusions

The influence of thyroid hormones on the proper functioning of the nervous system is beyond doubt. Hashimoto encephalopathy is a rare disease with nonspecific clinical manifestation. Despite its incidental occurrence, it should be considered in the differential diagnosis in patients with acute or subacute features of encephalopathy, as well as with progressive cognitive impairment and psychiatric symptoms of unclear etiology. Taking into account the possibility of the coexistence of various autoantibodies, HE should, in some cases, be differentiated from other autoimmune encephalopathies and paraneoplastic syndromes. If elevated titers of antibodies to thyroid antigens are then demonstrated, corticosteroid treatment should be considered as soon as possible. Future research should look for a specific marker for Hashimoto encephalopathy.

## Figures and Tables

**Figure 1 cells-11-02873-f001:**
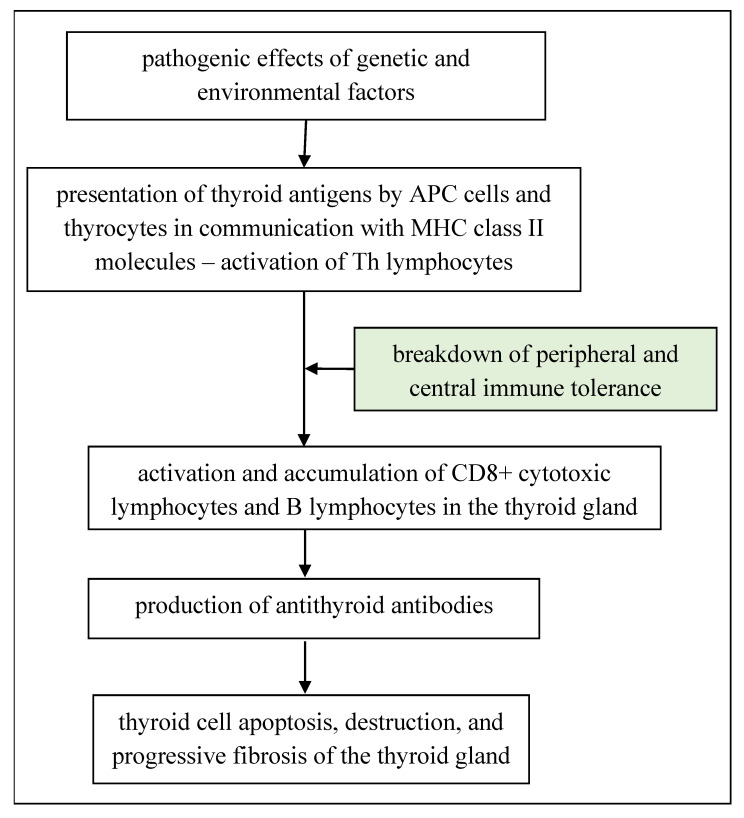
Diagram of the pathogenetic mechanism of Hashimoto’s thyroiditis, according to [4,38]—author’s own modification.

**Table 1 cells-11-02873-t001:** Clinical types of Hashimoto encephalopathy—according to [7,9,58,60]—author’s own modification.

Type 1—Recurrent, Benign, Vasculitic	Type 2—Progressive, Indolent
− suggesting a vascular background − transient ischemic attack (TIA) − stroke-like incidents − transient aphasia − focal deficit − psychoneurological manifestations—cognitive dysfunction − altered consciousness	− status epilepticus − disturbances of consciousness − psychiatric symptoms—mania, depression − psychotic disorders—paranoid, visual hallucinations, acute mood disturbances, delirious mania, catatonia − an increasing dementia syndrome
Symptoms common in both types	
− seizures − myoclonus − movement disorders − cerebellar manifestations—dysmetria, dysdiadochokinesia, opsoclonus − tremor − gait disorder or ataxia − stupor, coma	

**Table 2 cells-11-02873-t002:** Diagnostic criteria for Hashimoto encephalopathy (based on Graus et al. [106]).

1. Encephalopathy with seizures, myoclonus, hallucinations or stroke-like episodes
2. Thyroid disease (subclinical or mild overt)
3. MRI scan of brain—normal or with nonspecific abnormalities
4. Serum thyroid antibodies present (no specific disease—cut-off value)
5. Absence of other neuronal antibodies in serum or CSF
6. Exclusion of alternative causes of encephalopathy by differential diagnosis

## Data Availability

The data presented in this study are available upon request from the corresponding author. The data are not publicly available.

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
