# Peer review of "Hashimoto Encephalopathy—Still More Questions than Answers"

_cells, 2022, doi:10.3390/cells11182873_

Round 1
Reviewer 1 Report
Overall, this is an accurate, interesting and in-depth review of the recent literature about the thyroid–brain axis and the etiology, pathogenesis, and controversy surrounding Hashimoto encephalopathy, the most serious neurological complication of Hashimoto’s thyroiditis. Although the citations are numerous, relevant and updated, I would like the authors to deepen the pathogenesis of both Hashimoto's thyroiditis and Hashimoto encephalopathy by adding something about the role of T and B lymphocytes and explaining better what are the various pathogenetic steps that lead to these pathological conditions.
Author Response
Dear Editor,
Dear Reviewers,
Thank you for your efforts in processing our paper. We greatly appreciate reviewers’ thoughtful comments, which we believe have helped us to improve the paper substantially. Below we provide a point-by-point response to these comments. All changes in the manuscript have been marked with a track-changes mode.
On behalf of all authors,
Marta Waliszewska-ProsóÅ‚
Reviewer #1:
Point: Although the citations are numerous, relevant and updated, I would like the authors to deepen the pathogenesis of both Hashimoto's thyroiditis and Hashimoto encephalopathy by adding something about the role of T and B lymphocytes and explaining better what are the various pathogenetic steps that lead to these pathological conditions.
Response: This has been added to the text and in the Figure 1 as suggested.
Reviewer 2 Report
Overall, the aim of this review is unclear.
Methods
The author searched only for Hashimoto's encephalopathy and SERAT, while EAATD and NAIM are also the same disease concept. Many records may be underestimated.
Table1,
The classification of Hashimoto's encephalopathy into two types is interesting, but there is no evidence for the classification criteria. If the classification is what the authors wish to propose, they need to provide citations for the evidence.
Chapter 3.3.3. is titled Clinical Manifestations, but it also discusses laboratory and imaging findings.
Lin423, if imaging findings are to be positioned as important, the review should be written with that logic in mind, and other articles on imaging studies should be compared and evaluated.
We recommend that you write your review around the areas you want to focus on, rather than summarizing what has often been said in previous reviews.
Author Response
Dear Editor,
Dear Reviewers,
Thank you for your efforts in processing our paper. We greatly appreciate reviewers’ thoughtful comments, which we believe have helped us to improve the paper substantially. Below we provide a point-by-point response to these comments. All changes in the manuscript have been marked with a track-changes mode.
On behalf of all authors,
Marta Waliszewska-ProsóÅ‚
Reviewer #2:
Point: The author searched only for Hashimoto's encephalopathy and SERAT, while EAATD and NAIM are also the same disease concept. Many records may be underestimated.
Response: The terms EAATD and NAIM were also searched, our oversight in the methodology has already been corrected.
Point: Table1, The classification of Hashimoto's encephalopathy into two types is interesting, but there is no evidence for the classification criteria. If the classification is what the authors wish to propose, they need to provide citations for the evidence.
Response: The caption in the table description has been corrected that this is a conventional clinical division based on the literature as modified by the authors themselves.
Point: Chapter 3.3.3. is titled Clinical Manifestations, but it also discusses laboratory and imaging findings.
Response: Thank you for this comment. Actually, subsection 3.3.3. was imprecise. We have moved unnecessary information to other chapters.
Point: Lin423, if imaging findings are to be positioned as important, the review should be written with that logic in mind, and other articles on imaging studies should be compared and evaluated.
Response: We have changed the form in which this information is presented in the text.
Round 2
Reviewer 1 Report
The authors made the changes I suggested in a timely and thorough manner. The manuscript now is complete and exhaustive.
Reviewer 2 Report
The author has provided appropriate responses and edited the paper with regard to the points raised.
There are no additional remarks.